# Are Mutations in the DHRS9 Gene Causally Linked to Epilepsy? A Case Report

**DOI:** 10.3390/medicina56080387

**Published:** 2020-08-01

**Authors:** Francesco Calì, Maurizio Elia, Mirella Vinci, Luigi Vetri, Edvige Correnti, Emanuele Trapolino, Michele Roccella, Francesca Vanadia, Valentino Romano

**Affiliations:** 1Oasi Research Institute-IRCCS, 94018 Troina, Italy; cali@oasi.en.it (F.C.); mvinci@oasi.en.it (M.V.); luigi.vetri@gmail.com (L.V.); 2Department of Sciences for Health Promotion and Mother and Child Care G. D’Alessandro, University of Palermo, 90127 Palermo, Italy; edvigecorrenti@gmail.com; 3Child Neuropsichiatry Unity, ISMEP-G. di Cristina Hospital-Arnas Civico, 90134 Palermo, Italy; emanuele.trapolino@arnascivico.it (E.T.); francesca.vanadia@arnascivico.it (F.V.); 4Department of Psychology, Educational Science and Human Movement, University of Palermo, 90128 Palermo, Italy; michele.roccella@unipa.it; 5Department of Biological, Chemical and Pharmaceutical Sciences and Technologies, University of Palermo, 90128 Palermo, Italy; valentino.romano@unipa.it

**Keywords:** DHRS9, allopregnanolone, GABA, temporal lobe epilepsy, NGS, exome

## Abstract

The DHRS9 gene is involved in several pathways including the synthesis of allopregnanolone from progesterone. Allopregnanolone is a positive modulator of gamma aminobutyric acid (GABA) action and plays a role in the control of neuronal excitability and seizures. Whole-exome sequencing performed on a girl with an early onset epilepsy revealed that she was a compound heterozygote for two novel missense mutations of the DHRS9 gene likely to disrupt protein function. No previous studies have reported the implication of this gene in epilepsy. We discuss a new potential pathogenic mechanism underlying epilepsy in a child, due to a defective progesterone pathway.

## 1. Introduction

Genetic studies have revealed a role of the gamma-aminobutyric acid (GABA) system in the pathogenesis of epilepsy and experimental and clinical studies have highlighted the importance of GABA in the treatment of epilepsy [1]. Allopregnanolone (AP) and tetrahydrodeoxycorticosterone (THDOC) are positive allosteric modulators of GABA action at GABAA receptors and play a role in the control of neuronal excitability and seizures [2,3]. Allopregnanolone is synthesized in the nervous system from progesterone by the sequential action of two enzymes: 5α-reductase type I (5α-RI) (SRD5A1 gene), which transforms progesterone into 5α-dihydroprogesterone, and 3α-hydroxysteroiddehydrogenase (3α-HSD) (AKR1C2 gene) which converts 5α-dihydroprogesterone into allopregnanolone (3β-tetrahydroprogesterone) [4]. The back-oxidation of 3β-tetrahydroprogesterone to 5α-dihydroprogesterone is catalyzed by a NAD-dependent 3α-hydroxysteroid dehydrogenase (DHRS9 gene) [5] (Figure 1). Moreover, Birgit Stoffel-Wagner et al. [6] demonstrated that the 3a-HSD-1 mRNA was not expressed in the hippocampus and temporal lobe of patients with epilepsy.

## 2. Clinical Report

Here, we describe a 4-year-old girl showing a mild global development delay. The girl had a good adaptation to extrauterine life (APGAR score: 9–10). Her birth weight was 3260 g (Z-score −0.36), length 50 cm (Z-score −0.01), and head circumference 35 cm (Z-score 0.83). Seizures started at two months of life, several times a day, characterized by upward or rightward eye revulsion and immobility; in the following weeks, very frequent epileptic spasms occurred, most frequent on awakening and in falling asleep. Anti-epileptic therapy was administered with valproic acid, levetiracetam, pyridoxine supplementation and adrenocorticotrophic hormone treatment, but seizures were initially drug-resistant. The ictal electroencephalogram showed high-amplitude rhythmic epileptiform discharges at 9 Hz, followed by high-voltage generalized slow waves in the right temporal regions. The interictal EEG was also abnormal with a suppression-burst pattern. The brain magnetic resonance imaging performed at the same age revealed an enlargement of brain cerebrospinal fluid spaces and a thinning of the corpus callosum, without focal lesions. Metabolic investigations resulted normal. Seizures were controlled at 1 year of age, with valproic acid and clonazepam, and EEG did not show epileptiform abnormalities at the last visit, at 4 years of age.

This study was approved by the local Ethics Committee “Comitato Etico IRCCS Sicilia-Oasi Maria SS”, Prot. N. 2017/05/31/CE-IRCCS-OASI/9 as of 3 June 2017. The study was conducted in accordance with the Declaration of Helsinki. Written informed consent was obtained from the patient’s parents. All investigations were conducted according to the principles expressed in the Declaration of Helsinki. Nuclear DNA from the patient and her parents was isolated from peripheral blood leucocytes. Array-CGH was performed (resolution 400k), but no genomic imbalances were detected. Library preparation and exome enrichment were performed using the Agilent SureSelect V7 (Agilent Technologies, Santa Clara, CA, USA) kit according to the manufacturer’s instructions. The quality of post-amplification libraries was assessed using DNA 1000 chips on the BioAnalyzer 2100 (Agilent) and Qubit fluorimetric quantitation using Qubit dsDNA BR Assay Kits (Invitrogen, Carlsbad, CA, USA). An indexed 150 bp paired-end sequencing run was performed on an Illumina HiSeq 3000 instrument at the CRS4 NGS facility. This approach achieved a 75× average coverage over the 36 Mb of genomic regions sequenced, with 95% regions covered at least 20×. We filtered the identified variants according to: (i) recessive/de novo/X-linked pattern of inheritance, (ii) allele frequencies (Mean Average Frequency, MAF) <1% using as reference the following genomic datasets: 1000 Genomes, ESP6500, ExAC, gnomAD.

Exome sequencing performed on nuclear DNA isolated from the patient and her parents showed that she is a compound heterozygote for the c.785C > T (p.Ser262Leu) and c.1036G > C (p.Asp346His) missense mutations of the DHRS9 gene inherited from the mother and the father, respectively.

The two mutations caused changes in two highly conserved amino acid sites in the DHRS9 protein. Furthermore, the “in silico predictive tool” used (Table 1) suggested that the mutations were likely to have a deleterious/damaging effect on DHRS9 function. No other variant was identified in candidate epilepsy genes.

## 3. Discussion

Based on our findings, we propose that in this patient the epileptic seizures starting in the right temporal regions may be explained by the effect of the missense mutations on DHRS9 function. The absence or dysfunction of DHRS9 could block the back-oxidation of 3β-tetrahydroprogesterone to 5α-dihydroprogesterone (Figure 1). We speculate that in this patient seizures are due to an impairment of allopregnanolone level—due to the above described block—with a consequent effect on inhibitory gabaergic neurons. Stoffel-Wagner et al. [6] have reported the absence of 3a-HSD-1 mRNA in the hippocampus and in the temporal lobe of epileptic patients. Similarly, Lucchi et al. [7] have demonstrated, in their recent study, a significantly reduction in pregnenolone, allopregnanolone and pregnanolone in the hippocampus of epileptic rats. However, interestingly, they found a higher allopregnanolone synthesis in presence of an increased epileptic activity.

Moreover, there is literature evidence that a chronic exposure to increased levels of allopregnanolone (e.g., during stress, menstrual cycle, pregnancy) leads, with an unclear mechanism, to a decrease in the abundance of the GABA-A receptor α4 subunit and a decrease in the expression of the α4 subunit mRNA in the ventral-posteriomedial nucleus of the thalamus [8].

It thus appears that epileptogenesis may result either from an absence of, or a sustained increase in, allopregnanolone. A clear limit of our article is that we have no proof of changes in the brain allopregnanolone concentration. Therefore, more studies are needed on epileptic patients with a defective progesterone pathway, and the use of in vitro functional assays to knock-out the expression of DHRS9 or AKR1C2, such as CRISP-R/Cas 9 and siRNA technologies, will be required to test the validity of the proposed pathogenic model.

## Figures and Tables

**Figure 1 medicina-56-00387-f001:**
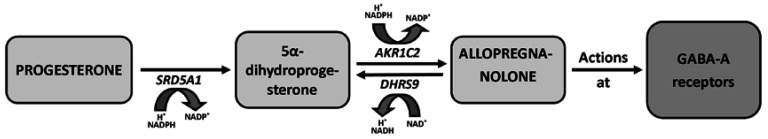
Biosynthetic pathway of the allopregnanolone from progesterone. The patient is a compound heterozygote for two mutations in the DHRS9, preventing the back oxidation of allopregnanolone to 5alpha-dihydroprogesterone (5alpha-DHP) (see text for details and discussion).

**Table 1 medicina-56-00387-t001:** In silico prediction of the effects of missense mutations, c.785C > T (p.Ser262Leu) and c.1036G > C (p.Asp346His) in the DHRS9 (NM_001289763) gene.

In Silico Predictive Tool	Prediction/Score		PHRED-Scaled
Mutation	c.785C > T (p.Ser262Leu)	c.1036G > C (p.Asp346His)	
dbSNP	rs776765324	rs11695788	
PA_CADD_phred	21.5	29.6	>30 highly pathogenic; >20 pathogenic
PA_DANN_score	0.996382	0.995699	range from 0 to 1 *
PA_Eigen-phred	1.87611	13.87712	
PA_FATHMM_pred	DAMAGING	DAMAGING	
PA_GERP++_RS	5.14	5.93	range from −12.3 to 6.17 *
PA_LRT_pred	Deleterious	Deleterious	
PA_M-CAP_pred	DAMAGING	DAMAGING	
PA_MetaSVM_pred	DAMAGING	DAMAGING	
PA_MutationAssessor_pred	MEDIUM	HIGHT	neutral, low, medium, high
PA_MutationTaster_pred	Polymorphism	DAMAGING	
PA_PROVEAN_pred	DAMAGING	DAMAGING	
PA_Polyphen2_HDIV_pred	Benign	Probably Damaging	
PA_SIFT_pred	Tolerated	Deleterious	
PA_SiPhy_29way_logOdds	16.9774	19.9359	range from 0 to 37.9718 *
PA_fathmm-MKL_coding_pred		DAMAGING	
ClinVar (Clinical Significance)	Not Reported in ClinVar	Not Reported in ClinVar	
Frequency (GnomAD)	0.000004	0.003797	

* A larger number indicates a higher probability to be damaging.

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
