# Peer review of "Are Mutations in the DHRS9 Gene Causally Linked to Epilepsy? A Case Report"

_medicina, 2020, doi:10.3390/medicina56080387_

Round 1

Reviewer 1 Report

The research is very interesting and promising, but right now, it is no more than a case report. Hence, the title must be rewritten to convey it. The general idea the authors are launching may be plausible but not licit from the scientific point of view; they need to present more results and less speculation. Authors have to demonstrate the increase of the allopregnanolone levels and what is more important: correlating it with the alteration of GABAergic system, before speculating about it and before mentioning “toxin effect on the inhibitory gabaergic neurons”.

Author Response

Dear reviewer,

I would like to thank the editor and the reviewers for your advice and corrections to the article. As you requested, we have made all necessary changes in our manuscript to address the editor and reviewers’ concerns and have detailed below how the points raised by the referees have been accommodated. The main changes are written in red in the text of the manuscript. From the changes made in the revised manuscript and responses provided below, I hope you are convinced that we have adequately addressed the reviewer’s concern and made the paper better. If there are any further questions, please feel free to let me know.

Reviewer 1

Point 1: The research is very interesting and promising, but right now, it is no more than a case report. Hence, the title must be rewritten to convey it.

Response 1: We have changed the title in “Are mutations in the DHRS9 gene causally linked to temporal lobe epilepsy? A case report.” in order to underline that the paper is essentially a case report.

Point 2: The general idea the authors are launching may be plausible but not licit from the scientific point of view; they need to present more results and less speculation. Authors have to demonstrate the increase of the allopregnanolone levels and what is more important: correlating it with the alteration of GABAergic system, before speculating about it and before mentioning “toxin effect on the inhibitory gabaergic neurons”.

Response 2: We agreewith your suggestions. We have increased the literature evidenceabout the implication of allopregnanolone levels on GABAergic systemin the Discussion. Moreover, we have added that the impossibility to prove a change in the brain allopregnanolone level is a limit of our study.

Sincerely,

Dr. Maurizio Elia

Troina, July 27, 2020

Reviewer 2 Report

The authors report a child with early onset epilepsy, at the age of two months, seizures of generalized pattern, including eye deviations, epileptic spasms, more generalized than focal. Not typical TLE.

Most frequent on awakening and in falling asleep.

The interictal EEG showed suppression burst pattern. The ictal electroencephalogram showed rhythmic epileptiform discharges at 9 Hz with main amplitude, main amplitude: what’s that? .. and followed by high-voltage generalized slow waves in the right temporal lobe. This seems to be slow waves without focal importance, should be commented.

This girl had enlarged ventricles and a thin corpus callosum indicating some syndrome.

It is absolutely of interest that the genetic defect may lower the threshold for epilepsy in the patient. Did they try other specific GABA-A agonists? They refer to Stoffel-Wagnera; her name is Birgit Stoffel-Wagner! They demonstrated that mRNA expression of 5 alpha-reductase 1 and 3 alpha-HSD 2 and 3 and 20 alpha-HSD in the hippocampus and temporal lobe of epileptic patients; probably not only in TLE.

The authors are using TLE in this case. It is not typical TLE and not in a small child. They better us epilepsy only.

Author Response

Dear reviewer,

I would like to thank the editor and the reviewers for your advice and corrections to the article. As you requested, we have made all necessary changes in our manuscript to address the editor and reviewers’ concerns and have detailed below how the points raised by the referees have been accommodated. The main changes are written in red in the text of the manuscript. From the changes made in the revised manuscript and responses provided below, I hope you are convinced that we have adequately addressed the reviewer’s concern and made the paper better. If there are any further questions, please feel free to let me know.

Reviewer 2

Point 1: The authors report a child with early onset epilepsy, at the age of two months, seizures of generalized pattern, including eye deviations, epileptic spasms, more generalized than focal. Not typical TLE.

Response 1: Thanks to your suggestions, we did not refer to the epilepsy of our patient as a TLE.

Point 2: Most frequent on awakening and in falling asleep.

Response 2: Many thanks, we rectified.

Point 3: The interictal EEG showed suppression burst pattern. The ictal electroencephalogram showed rhythmic epileptiform discharges at 9 Hz with main amplitude, main amplitude: what’s that? .. and followed by high-voltage generalized slow waves in the right temporal lobe. This seems to be slow waves without focal importance, should be commented.

Response 3: We have better clarified that both epileptiform discharges and subsequentgeneralized slow waves are in the right temporal lobe region.

Point 4: This girl had enlarged ventricles and a thin corpus callosum indicating some syndrome.

Response 4: We have no other elements indicating some specific syndrome. We could speculate that the dysregulation of allopregnanolone level altersGABAergic system with unclear consequences in early brain development.

Point 5: It is absolutely of interest that the genetic defect may lower the threshold for epilepsy in the patient. Did they try other specific GABA-A agonists?

Response 5: Yes, we have tried clonazepam (line 66).

Point 6: They refer to Stoffel-Wagnera; her name is Birgit Stoffel-Wagner! They demonstrated that mRNA expression of 5 alpha-reductase 1 and 3 alpha-HSD 2 and 3 and 20 alpha-HSD in the hippocampus and temporal lobe of epileptic patients; probably not only in TLE.

The authors are using TLE in this case. It is not typical TLE and not in a small child. They better us epilepsy only.

Response 6:We have appreciated your suggestions and we have corrected as above indicated.

Sincerely,

Dr. Maurizio Elia

Troina, July 27, 2020

Reviewer 3 Report

More elaborate discussion about toxic effect of allopregnanolone on GABAergic neurons could be useful. 

Both acute and chronic tolerances can develop to the effects of allopregnanolone, but the exact mechanisms are unknown.

(Sahruh Turkmen, Torbjorn Backstrom, Goran Wahlstrom, Lotta Andreen and Inga-Maj Johansson -  “Tolerance to allopregnanolone with focus on the GABA-A receptor´, British Journal of Pharmacology (2011) 162 311–327 311)

Author Response

Dear reviewer,

I would like to thank the editor and the reviewers for your advice and corrections to the article. As you requested, we have made all necessary changes in our manuscript to address the editor and reviewers’ concerns and have detailed below how the points raised by the referees have been accommodated. The main changes are written in red in the text of the manuscript. From the changes made in the revised manuscript and responses provided below, I hope you are convinced that we have adequately addressed the reviewer’s concern and made the paper better. If there are any further questions, please feel free to let me know.

Reviewer 3

Point 1: More elaborate discussion about toxic effect of allopregnanolone on GABAergic neurons could be useful. 

Response 1: We agreewith your suggestions. We have increased the literature evidence about the implication of allopregnanolone levels on GABAergic systemin the discussion.

Point 2: Both acute and chronic tolerances can develop to the effects of allopregnanolone, but the exact mechanisms are unknown.

(Sahruh Turkmen, Torbjorn Backstrom, Goran Wahlstrom, Lotta Andreen and Inga-Maj Johansson -  “Tolerance to allopregnanolone with focus on the GABA-A receptor´, British Journal of Pharmacology (2011) 162 311–327 311)

Response 2: Many thanks for your suggestionwe have added this evidence in the Discussion.

Sincerely,

Dr. Maurizio Elia

Troina, July 27, 2020

Round 2

Reviewer 1 Report

I fully agree with the changes performed and have no objection.